# Generating Images with Multimodal Language Models

**Jing Yu Koh**
Carnegie Mellon University
jingyuk@cs.cmu.edu

**Daniel Fried**
Carnegie Mellon University
dfried@cs.cmu.edu

**Ruslan Salakhutdinov**
Carnegie Mellon University
rsalakhu@cs.cmu.edu

## Abstract

We propose a method to fuse frozen text-only large language models (LLMs) with pre-trained image encoder and decoder models, by mapping between their embedding spaces. Our model demonstrates a wide suite of multimodal capabilities: image retrieval, novel image generation, and multimodal dialogue. Ours is the first approach capable of conditioning on arbitrarily interleaved image and text inputs to generate coherent image (and text) outputs. To achieve strong performance on image generation, we propose an efficient mapping network to ground the LLM to an off-the-shelf text-to-image generation model. This mapping network translates hidden representations of text into the embedding space of the visual models, enabling us to leverage the strong text representations of the LLM for visual outputs. Our approach outperforms baseline generation models on tasks with longer and more complex language. In addition to novel image generation, our model is also capable of image retrieval from a prespecified dataset, and decides whether to retrieve or generate at inference time. This is done with a learnt decision module which conditions on the hidden representations of the LLM. Our model exhibits a wider range of capabilities compared to prior multimodal language models. It can process image-and-text inputs, and produce retrieved images, generated images, and generated text — outperforming non-LLM based generation models across several text-to-image tasks that measure context dependence.

## 1   Introduction

Autoregressive language models (LMs) and large language models (LLMs) trained on text corpora have shown impressive abilities to efficiently adapt to other modalities. Prior work showcased the effectiveness of grounding text-only LMs to images for vision-and-language tasks [56, 4, 29, 33, 31, 35], to embodied settings for robotics [3, 18], offline reinforcement learning [48], and more. These methods typically keep most of the LLM weights frozen. This allows them to leverage the capabilities that the LLM learns during large scale text-only pretraining, such as the ability to learn from in-context examples [9], more effectively process longer context, and condition on inputs more strongly.

In this work, we tackle the task of extending multimodal language models to generate novel images. Our approach, **G**enerating **I**mages with **L**arge **L**anguage Models (GILL), is capable of processing arbitrarily interleaved image-and-text inputs to generate text, retrieve images, and generate novel images (Fig. 1). Our findings show that it is possible to efficiently map the output embedding space of a frozen text-only LLM to that of a frozen generation model (in this work, Stable Diffusion [49]) despite both models using entirely different text encoders. We achieve this by finetuning a small number of parameters on image-caption pairs [52], in contrast to other methods which require interleaved image-text data [4, 2]. Our approach is computationally efficient and does not require running the image generation model at training time. To achieve strong image generation performance, we propose efficient architectural changes to learn the LLM-to-generation mapping effectively with the GILLMapper module. GILLMapper is a lightweight Transformer [57] conditioned on special

37th Conference on Neural Information Processing Systems (NeurIPS 2023).

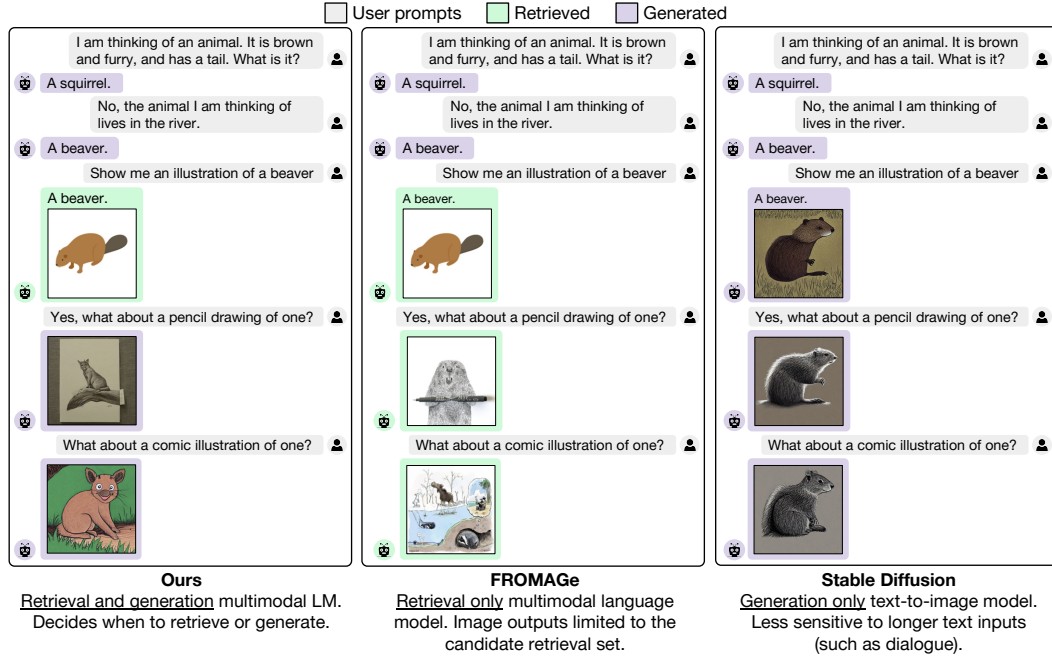

Figure 1: Our model is capable of generating text, retrieving images, generating novel images, and interleaving results into coherent multimodal dialogue.

learnt text tokens. We train it by minimizing the $l_2$ distance between its outputs and the outputs of the text encoder of a text-to-image generation model. This distillation training allows us to use the image decoder of the text-to-image model at inference time. Despite its simplicity, we show that this allows us to outperform the baseline text-to-image generation model on several tasks that measure language context dependence. Finally, to decide whether to produce a retrieved image or a generated one at inference time, we train a decision model that outputs a decision conditioned on the LM hidden representations. This allows us to both generate and retrieve in output sequences, as shown in Fig. 1.

Our experimental results demonstrate that GILL is more effective than Stable Diffusion at processing longer-form text, including dialogue and discourse. We show on dialogue-conditioned image generation that GILL can outperform non-LLM based generation models, and benefit from multimodal context: generating images that match text *better* than the backbone generation models that we distill from. In addition, GILL can process arbitrarily interleaved image-text inputs, unlike typical text-to-image models which only process text. GILL is the first model capable of outputting retrieved images, novel images, and text — interleaving these for coherent multimodal dialogue generation.[1]

## 2 Related Work

**Multimodal Language Models**    Several prior works have developed multimodal language models which process image and text inputs to generate text outputs. Frozen [56] showed that it is possible to finetune a visual encoder to map images into the hidden space of a text-only LLM, and that this exhibits compelling few-shot, captioning, and question answering abilities. Other methods improve upon this approach by introducing adapters [19], scaling up model and data sizes [4, 64], improving the visual encoder [4, 33], finetuning on instructions [35], and training unified models on multi-task objectives [36, 63, 42]. CM3 [2, 62] trained multimodal LMs on HTML webpages consisting of interleaved images and text. Many state-of-the-art models also require significant computational resources to train. For example, Flamingo [4] is trained on 1535 TPUs for 15 days, while RA-CM3 [62] use 256 GPUs for 5 days. In contrast, our efficient adaptation method is trained on 2 GPUs for 2 days. The most similar work to our approach is FROMAGe [31], which trains a multimodal language model capable of processing arbitrarily interleaved image and text inputs to

---

[1]Our code and pretrained models are publicly released at `https://github.com/kohjingyu/gill`.

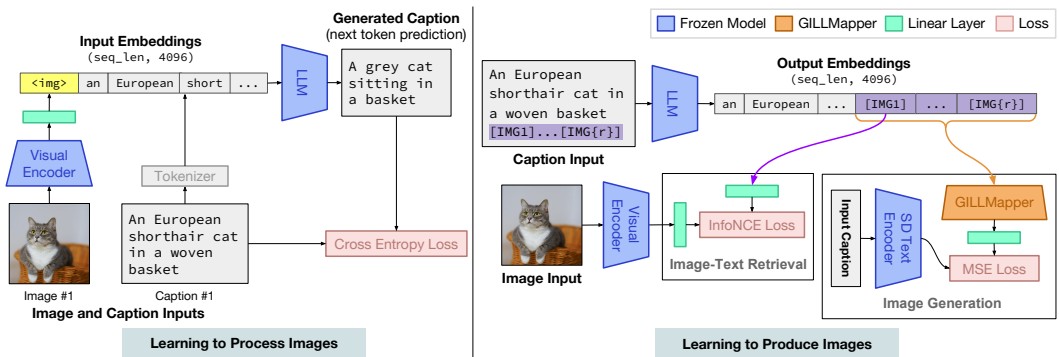

Figure 2: GILL model architecture overview. It is trained with a captioning loss to learn to process images (left), and losses for image retrieval and image generation to learn to produce images (right).

generate text interleaved with retrieved images. While FROMAGe can only retrieve images in their outputs, GILL is capable of both image retrieval and image generation, which allows it to outperform retrieval-only models when they are limited by their candidate retrieval set (Fig. 5).

**Large Language Models** Our work leverages recent advances in Transformer-based [57] LLMs. When trained at large enough scale, LLMs exhibit compelling properties, such as the ability to learn from few-shot in-context examples [9, 11] and generate and process long text inputs [61, 59, 53, 7]. Our approach also builds upon recent efforts on open sourced LLM weights [69, 55].

**Text-to-Image Generation** Text-to-image generation is the task of synthesizing a realistic image conditioned on natural language descriptions. [47] was one of the first to tackle this with a conditional GAN [23]. Later work improved upon this by introducing multi-stage models [67], attention mechanisms [60], and contrastive methods [73, 66]. Several recent papers also formulate the text-to-image generation task as a sequence modeling problem [45, 17, 13], training large Transformer [57] models on discretized image tokens [46]. [20, 65] improved upon this approach by introducing stronger image quantizers and scaling up model parameters. Several recent methods [38, 44, 49] apply diffusion models [26] to improve generated image quality. [50, 65] scale up text encoder models to achieve significant gains in generating relevant images. In contrast with computationally intensive methods that train end-to-end, GILL does not require running the image generation model during training.

## 3 Method

We efficiently adapt a pretrained autoregressive language model of text, to *process* image and text inputs and *produce* image and text outputs. Most of the model weights (including those of the base LLM and image generator) are kept frozen, and we finetune a small number of parameters on image caption data (Fig. 2) to achieve a wide range of capabilities (Fig. 5). There are several challenges that need to be resolved. The model needs to learn to process image-and-text content (Sec. 3.1). It also needs to learn to produce images (either retrieved or generated), and determine whether to produce text or images at each step (Sec. 3.2). Finally, whenever an image is produced, the model needs to decide whether image retrieval (from a candidate set) or generation is more appropriate (Sec. 3.3).

### 3.1 Learning to Process Images

Given an image $x$ and its text caption $y$ (tokenized as $(s_1, \ldots, s_T)$), our goal is to adapt a frozen LLM to enable it to complete any sequence of arbitrarily interleaved image and text inputs. For example, inputs for the Visual Storytelling dataset [28] consist of 5 images and 5 text descriptions, interleaved in a manner such as $(x_1, y_1, \ldots, x_5, y_5)$. We follow prior work [56, 19, 35, 31] in learning translation parameters that map from image features to text embedding space.

We first extract visual embeddings $v_\phi(x) \in \mathbb{R}^d$ with a pretrained visual backbone (its weights $\phi$ and LLM weights $\theta$ are kept frozen). We learn a linear mapping $\mathbf{W}_{\text{cap}} \in \mathbb{R}^{d \times ke}$ which maps $v_\phi(x)$ into a sequence of $k$ $e$-dimensional vectors that we use as inputs to the LLM (Fig. 2, left), where $e$ is the

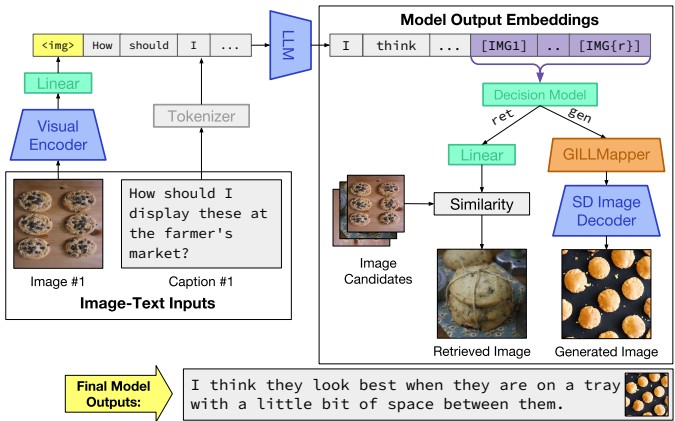

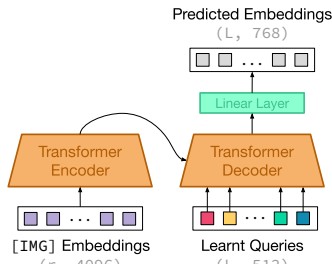

Figure 4: GILLMapper model architecture. It is conditioned on the hidden [IMG] representations and a sequence of learnt query embedding vectors.

Figure 3: Inference time procedure for GILL. The model takes in image and text inputs, and produces text interleaved with image embeddings. After deciding whether to retrieve or generate for a particular set of tokens, it returns the appropriate image outputs.

LLM input embedding dimension. We train $\mathbf{W}_{\text{cap}}$ on image-caption pairs (details in Sec. 3.4), by minimizing the negative log-likelihood loss of the token sequence $(s_1, \ldots, s_T)$:

$$l_c(x, y) = -\sum_{t=1}^{T} \log p_\theta(s_t \mid v_\phi(x)^T \mathbf{W}_{\text{cap}}, s_1, \ldots, s_{t-1}) \tag{1}$$

Intuitively, this objective trains a mapping $\mathbf{W}_{\text{cap}}$ that allows us to translate images into embedding vectors in the token embedding space of the LLM (illustrated in Fig. 2, left).

### 3.2 Learning to Produce Images

In order to enable the model to produce image outputs, we add special [IMG] tokens to the vocabulary of the LLM, similar to [71, 31] which introduce special tokens correspond to images that should be output by the model. The hidden states that the LLM produces for these tokens will be used to retrieve or generate images. While [31] use a single token for their image retrieval model, we observed in our experiments that image generation requires much more finegrained textual information (Sec. 5). In order to improve the expressivity of the frozen LLM for novel image generation, we generalize to use $r$ tokens [IMG1], ..., [IMG{r}] for representing visual outputs.

Concretely, we add a trainable matrix $\mathbf{E}_{\text{img}} \in \mathbb{R}^{r \times e}$ to the embedding matrix of the frozen LLM, which represents the $r$ [IMG] token embeddings. We wish to train the model to learn *when* it should produce [IMG] tokens. This is done by minimizing the negative log-likelihood of producing the first [IMG] token conditioned on previously generated tokens:

$$l_p(y) = -\log p_{\{\theta \cup \mathbf{E}_{\text{img}}\}}(\texttt{[IMG1]} \mid s_1, \ldots, s_t) \tag{2}$$

The LLM weights $\theta$ are kept frozen, and we only update $\mathbf{E}_{\text{img}}$. During inference, we always generate the [IMG2], ..., [IMG{r}] tokens whenever the first [IMG1] token is produced. During training, the [IMG] tokens are appended to each caption (Fig. 2). The LLM hidden states of the [IMG] tokens are used for image retrieval and generation, as described in the following sections.

**Novel Image Generation**    In order for our LLM to produce image outputs, the [IMG] tokens need to be mapped into a semantically meaningful region of the input space of an image generation model $G_\psi$ (such as that of the Stable Diffusion [49] image decoder). In initial experiments, we found that training a simple linear mapping such as those used in previous work on retrieval [31] was insufficient, and that such a model was unable to handle more complex prompts (see Sec. 5 for analysis). Hence, we propose GILLMapper (Fig. 4), a lightweight 4-layer encoder-decoder transformer model with trainable weights $\omega$. The GILLMapper module $f_\omega$ conditions on $h_{\{\theta \cup \mathbf{E}_{\text{img}}\}}(y, \texttt{[IMG]})$ (the [IMG] representations from the last hidden layer of the LLM) and $L$ learnt query embeddings $(q_1, \ldots, q_L) \in \mathbb{R}^{L \times m}$ (where $L$ is the maximum input sequence length of the text-to-image generation backbone $G_\psi$).

The purpose of introducing learnable query embeddings is to enable GILLMapper to extract sequences of $L$ features from the LLM [IMG] hidden states. This is similar to the queries introduced in DETR [10] for object detection and BLIP-2 [33] for extracting image features. We optimize the GILL trainable weights ($q_1, \ldots, q_L$ and $\omega$) by minimizing the MSE loss of the GILLMapper model outputs against the embeddings produced by the text encoder ($T_\psi$) of a frozen text-to-image generation model:

$$l_g(y) = \| f_\omega \left( h_{\{\theta \cup \mathbf{E}_{\text{img}}\}}(y, \texttt{[IMG\{1\}]}), \ldots, h_{\{\theta \cup \mathbf{E}_{\text{img}}\}}(y, \texttt{[IMG\{r\}]}), q_1, \ldots, q_L \right) - T_\psi(y) \|_2^2 \quad (3)$$

This is essentially distilling from $T_\psi$ to learn a valid mapping from the output representations of our frozen LLM to the input space of $G_\psi$. Note that this does not require $G_\psi$ during training, so we can precompute $T_\psi(y)$ ahead of time, making training highly efficient. During inference, when [IMG] tokens are generated, we can synthesize an image by applying GILLMapper and the decoder $G_\psi$:

$$\text{Generated Image} = G_\psi(f_\omega(h_{\{\theta \cup \mathbf{E}_{\text{img}}\}}(y, \texttt{[IMG\{1\}]}), \ldots, h_{\{\theta \cup \mathbf{E}_{\text{img}}\}}(y, \texttt{[IMG\{r\}]}), q_1, \ldots, q_L))$$

where $h_{\{\theta \cup \mathbf{E}_{\text{img}}\}}(y, \texttt{[IMG\{i\}]})$ represents the hidden states from the last hidden layer of the modified LLM corresponding to the $i^{th}$ [IMG] token. The learnt query embeddings ($q_1, \ldots, q_L$) are part of the GILLMapper model weights, and are hence kept fixed during inference.

**Image Retrieval**   Similar to [31], we learn a linear mapping $\mathbf{W}_{\text{t2i}} \in \mathbb{R}^{e \times p}$ that maps the first token ([IMG1] to a $p$-dimensional vector. We also learn a linear mapping $\mathbf{W}_{\text{i2t}} \in \mathbb{R}^{d \times p}$ that map the pooled visual output of the image encoder $v_\phi(x)$ to a $p$-dimensional space. These represent image and text embeddings, and we train the model by minimizing the InfoNCE loss [39]:

$$l_r(\mathbf{x}_i, \mathbf{y}_i) = -\log \frac{\exp(\text{sim}(\mathbf{x}_i, \mathbf{y}_i, \mathbf{W}_{\text{t2i}})/\tau)}{\sum_{j=1}^N \exp(\text{sim}(\mathbf{x}_j, \mathbf{y}_i, \mathbf{W}_{\text{t2i}})/\tau)} - \log \frac{\exp(\text{sim}(\mathbf{x}_i, \mathbf{y}_i, \mathbf{W}_{\text{i2t}}))/\tau)}{\sum_{j=1}^N \exp(\text{sim}(\mathbf{x}_i, \mathbf{y}_j, \mathbf{W}_{\text{i2t}}))/\tau)} \quad (4)$$

where the similarity is computed as

$$\text{sim}(x, y, \mathbf{W}) = \frac{\left(\mathbf{W}^T v_\phi(x)\right)^T \left(\mathbf{W}^T h_{\{\theta \cup \mathbf{E}_{\text{img}}\}}(y, \texttt{[IMG1]})\right)}{\left\| \mathbf{W}^T v_\phi(x) \right\| \left\| \mathbf{W}^T h_{\{\theta \cup \mathbf{E}_{\text{img}}\}}(y, \texttt{[IMG1]}) \right\|}$$

During inference, we follow standard procedure [43] in retrieving the image with the highest cosine similarity (between image embeddings and the [IMG] tokens) from a candidate set of images.

### 3.3   Deciding to Generate or Retrieve

While learning to produce [IMG] tokens allows us to decide *when* to interleave images in text , the task of deciding whether to retrieve *or* generate from [IMG] tokens remains. Intuitively, for a given prompt, we would like to retrieve when there is a strong match from our set of candidate images, and generate otherwise. In order to evaluate this, we collect human annotations on PartiPrompts (P2) [65], a collection of prompts used to benchmark image generation models. P2 contains some prompts that are well-represented by naturally occurring images, but others that are unlikely to occur in natural image sets, making it a test of generative models. For each of the 1,632 examples in P2, we generate an image with the text-to-image generation model $G_\psi$, and use the CLIP ViT-L [43] model to retrieve the top ranked image from CC3M [52] according to the cosine similarity of image embeddings $v_\phi$.

We have 5 independent human annotators (details in the appendix) select which of the two images for each prompt, retrieved or generated, is better matched to the prompt. We labeled the examples where the generated image was selected as 'gen' (indicating prompts which we should generate an image for) and 'ret' for prompts that should have an image retrieved. We extract the most confident set of these annotations (retaining roughly 900 examples with an inter-annotator agreement of at least 4/5), and split them into a 67% train (600) and 33% test (300) split. We use this to train a linear classifier on the LLM [IMG] hidden states as a decision model for deciding when to retrieve or generate (more details and baselines are provided in the appendix). Although these annotations of retrieving versus generating are somewhat model dependent, we believe that this data is still a valuable metric during model development. We will release our annotations to encourage future work in this space.

## 3.4 Data and Implementation Details

The final training objective for a batch of image-text pairs $(\mathbf{x}, \mathbf{y})$ is the sum of the captioning loss $l_c$ (Eq. 1), image token prediction loss $l_p$ (Eq. 2), generation loss $l_g$ (Eq. 3) and retrieval loss $l_r$ (Eq. 4):

$$\min_{\mathbf{W}_{i2t}, \mathbf{W}_{t2i}, \mathbf{W}_{cap}, \mathbf{E}_{img}, \omega, q_{1:L}} \frac{1}{N} \sum_{i=1}^{N} \left( l_c(\mathbf{x}_i, \mathbf{y}_i) + l_p(\mathbf{y}_i) + l_g(\mathbf{y}_i) + l_r(\mathbf{x}_i, \mathbf{y}_i) \right) \tag{5}$$

The decision model is trained separately after convergence of the other components. The multitask loss (Eq. 5) trains GILL to process images ($l_c$), produce `[IMG]` tokens ($l_p$), generate images ($l_g$), and retrieve images ($l_r$). This enables it to generalize to a wide range of vision-and-language tasks.

We train on Conceptual Captions (CC3M) [52], which consists of 3.3M image-text pairs. Following [31], we pack two random examples together during training with probability 0.5 (i.e., 50% of the time, the input is a single image and caption example, while the other 50% of the time the input consists of a sequence consisting of two interleaved images and captions). We use the OPT-6.7B [69] model as the LLM backbone (which produce hidden states $h_\theta$ with embedding dim $e = 4096$). For the visual model used to extract features $v_\phi$ for captioning and retrieval, we use the CLIP [43] ViT-L model. For our text-to-image generation backbone $G_\psi$, we use the Stable Diffusion [49] v1.5 model (with $L = 77$ input vectors).[2] We use $k = 4$ visual tokens, and $r = 8$ learnt `[IMG]` tokens. We set the GILLMapper query embedding dimension $m = 512$. For retrieval, we use an embedding dimension $p = 256$. All pretrained model weights are kept frozen, and we only train the linear layers $\mathbf{W}_{i2t}$, $\mathbf{W}_{t2i}$, $\mathbf{W}_{cap}$, the `[IMG]` embedding matrix $\mathbf{E}_{img}$, and the GILLMapper parameters $\omega$ and query vectors $q_{1:L}$. In total, there are 50M trainable parameters, significantly fewer than in the frozen LLM and visual models (which total approximately 8B parameters). We use bfloat16 precision [1], and optimize using Adam [30] ($\beta_1 = 0.9$, $\beta_2 = 0.95$) with a learning rate of 0.001. We train with a batch size of 200 for 20K iterations, which takes 2 days on 2 A6000 GPUs. We follow [31] and concatenate captions to encourage the model to attend to relevant images within an image-text sequence.

# 4 Experiments

GILL is the first multimodal language model capable of conditioning on image-and-text inputs to generate meaningful images interleaved with text. Hence, our experiments primarily focus on evaluating its ability to produce novel images (Sec. 4.1). Our results show that GILL improves over Stable Diffusion [49] on tasks that require processing long-form text such as dialogue and discourse. We also benchmark the performance of models in deciding whether to retrieve or generate (see appendix). GILL is capable of generating text, retrieving images, and generating images. Despite being more general than prior work [56, 4, 31], we find that GILL performs comparably to or better than existing multimodal LMs on contextual image retrieval and text generation tasks (see Sec. 5).

## 4.1 Contextual Image Generation

To test the ability of our model against baseline methods for novel image generation, we run experiments on the VIST [28] and VisDial [16] datasets. These are the same datasets used in prior work [31] for benchmarking image retrieval conditioned on multimodal text-and-image context.

**Evaluation Metrics** The focus of our evaluation is on the ability of generative models to handle complex language descriptions. Hence, we compute metrics which measure the relevance of the generated image content. We evaluate models with two metrics:

1. **CLIP Similarity:** We use the CLIP [43] ViT-L image encoder to produce pooled representations of generated images and the corresponding real images, and report their cosine similarity. A higher score indicates that a generated image is more similar to the real image.

2. **Learned Perceptual Image Patch Similarity (LPIPS):** LPIPS [68] evaluates the distance between image patches. We measure LPIPS between real and generated images. A lower value indicates that two images are closer in perceptual space (i.e., more similar), while a higher value indicates that two images are more dissimilar.

---

[2]These models were selected for their strong performance and open source availability, but in principle our approach can be applied with any other pretrained models with similar function calls.

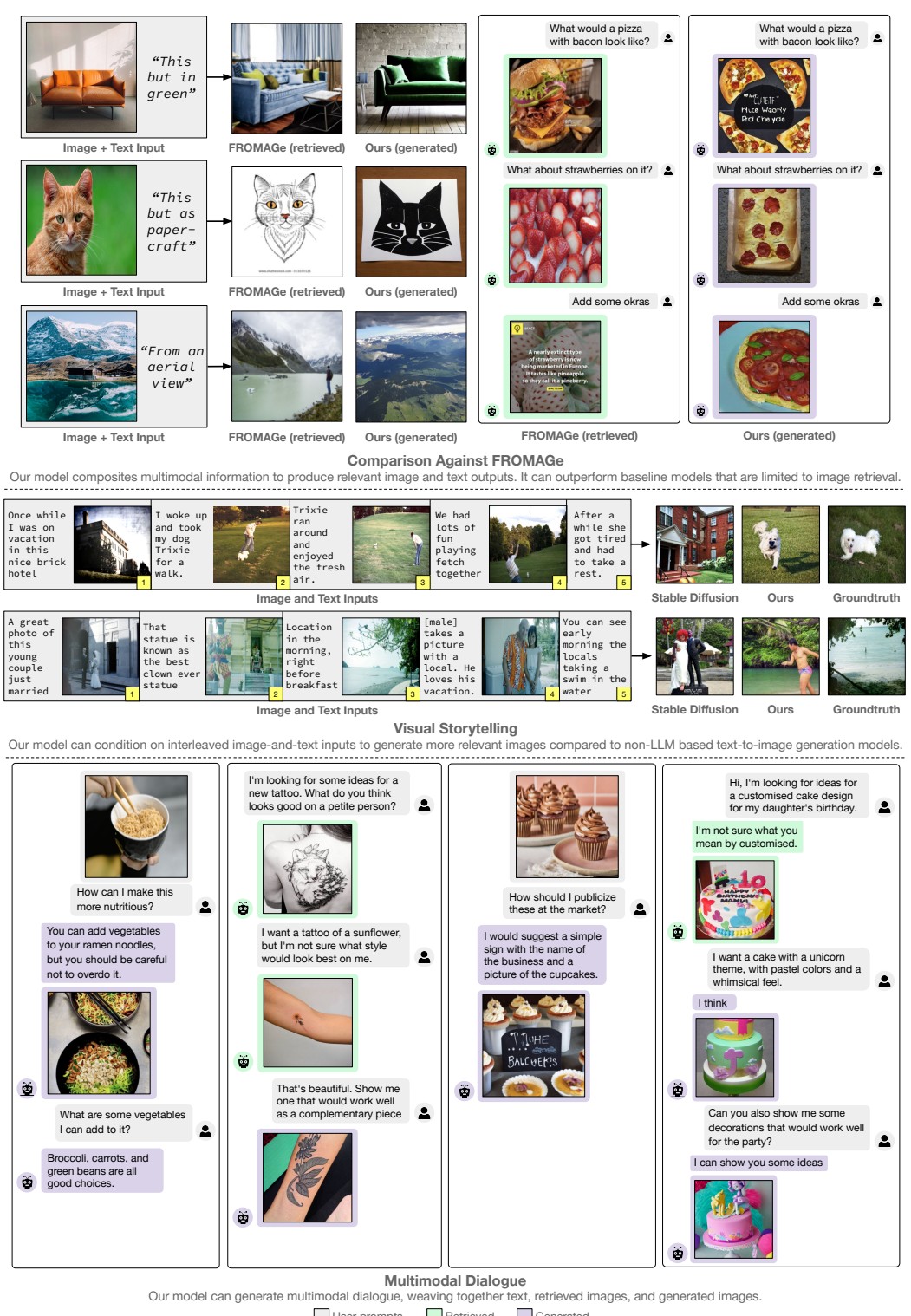

Figure 5: Qualitative results over various input and output modalities. GILL is able to process contextual multimodal cues to retrieve and generate appropriate image and text outputs.

Table 1: Results on contextual image generation on VIST [28] (averaged over 5 random seeds). Our model can process longer (possibly multimodel) inputs to outperform baseline models.

| Model | CLIP Similarity (↑) | | | LPIPS (↓) | | |
|---|---|---|---|---|---|---|
| | 1 caption | 5 captions | 5 caps, 4 images | 1 caption | 5 captions | 5 caps, 4 images |
| GLIDE [38] | 0.582 | 0.591 | - | 0.753 | 0.745 | - |
| Stable Diffusion [49] | **0.592** ±0.0007 | 0.598 ± 0.0006 | - | 0.703 ±0.0003 | 0.704 ± 0.0004 | - |
| GILL (ours) | 0.581 ±0.0005 | **0.612** ±0.0011 | **0.641** ±0.0011 | **0.702** ±0.0004 | **0.696** ±0.0008 | **0.693** ±0.0008 |

Table 2: Results on contextual image generation on VisDial [16] (averaged over 5 random seeds). Our model can process longer sequences of dialogue-like text to generate more relevant images.

| Model | CLIP Similarity (↑) | | | LPIPS (↓) | | |
|---|---|---|---|---|---|---|
| | 1 round | 5 rounds | 10 rounds | 1 round | 5 rounds | 10 rounds |
| GLIDE [38] | **0.562** | 0.595 | 0.587 | 0.800 | 0.794 | 0.799 |
| Stable Diffusion [49] | 0.552 ±0.0015 | **0.629** ±0.0015 | 0.622 ±0.0012 | **0.742** ±0.0010 | 0.722 ±0.0012 | 0.723 ±0.0008 |
| GILL (ours) | 0.528 ±0.0014 | 0.621 ±0.0009 | **0.645** ±0.0010 | **0.742** ±0.0022 | **0.718** ±0.0028 | **0.714** ±0.0006 |

**Generating from Visual Stories**    VIST [28] is a dataset for sequential vision-and-language tasks, with examples of sequences of 5 images and text that constitute a story, as shown in Fig. 5. Similar to [31], we test the models on generating the last image in the sequence, conditioned on different inputs:

1. **1 caption**: Input consists of the **last text description**. This is similar to standard text-to-image generation, where a model conditions on a single caption to generate an image.

2. **5 captions**: Input consists of all text from the **entire story sequence**. This tests the ability of models to process longer and temporally dependent text descriptions.

3. **5 captions, 4 images**: Lastly, we test models with inputs of **all images and texts preceding** the last image (i.e., sequenced as "`<text1><img1>`...`<text4><img4><text5>`"). This tests the ability of models to effectively process *multimodal context* in image generation. A novel feature of GILL is its ability to process interleaved image-text inputs, which most existing text-to-image generation models are unable to handle.

We report results on VIST in Tab. 1, comparing GILL against text-to-image generation baselines (including Stable Diffusion (SD) [49], which we use as our generation backbone $G_\psi$). With a single story caption input to both models, the performance is comparable, with SD achieving a slightly better CLIP Similarity score, and both models achieving similar LPIPS. However, when all 5 story captions are provided as input, our model outperforms SD, improving CLIP Similarity from 0.598 to 0.612, and LPIPS from 0.704 to 0.696. Interestingly, when further provided with the full multimodal context (the preceding 5 captions and 4 images), our model improves substantially, attaining a CLIP Similarity of 0.641 and LPIPS of 0.693. In contrast, SD is unable to handle interleaved image-text inputs without significant modifications. We also show several qualitative examples in Fig. 5. We find that GILL is generally more sensitive to input context compared to SD. GILL can also condition on image inputs, enabling it to use visual context to produce more relevant images.

We highlight that both models use the same image generation backbone, and the primary difference is in their text encoders. GILL is able to better handle long text inputs and multimodal context, which we attribute to the stronger LLM encoder coupled with our GILLMapper model.

**Generating from Visual Dialogue**    We also test our model on the VisDial [16] dataset. VisDial examples contain a sequence of question and answer (Q&A) pairs about a particular image, simulating dialogue between two people who are discussing an image. Examples contain up to 10 rounds of Q&A dialogue pairs. Similar to VIST, we evaluate the ability of models to accurately synthesize the image being described, provided with increasing amounts of the Q&A dialogue context as input. This experiment tests the ability of our approach to (1) generalize to dialogue-like text (as our approach is only finetuned on image caption data), and (2) process long text sequences.

Our results are presented in Tab. 2. Similar to the VIST evaluations, we find that with shorter length inputs, SD outperforms our model. However, when the input context is increased, our model gradually improves, and can synthesize images that are more similar to the groundtruth image. When the full 10 rounds of dialogue are provided, GILL significantly outperforms SD, improving over both CLIP

Table 3: Image generation performance on CC3M [52] and VIST [28] with different text mapping networks.

| | CC3M | VIST |
|---|---|---|
| **Model** | **FID** ($\downarrow$) | **CLIP Sim** ($\uparrow$) |
| Stable Diffusion [49] | **13.94** | 0.598 |
| Ours + Linear | 15.50 | 0.500 |
| Ours + 3-layer MLP | 15.33 | 0.502 |
| Ours + Transformer Encoder | 16.30 | 0.605 |
| Ours + GILLMapper | 15.31 | **0.641** |

Table 4: GILL image generation results on CC3M [52] with different number of image tokens ($r$).

| | CC3M | VIST |
|---|---|---|
| $r$ | **FID** ($\downarrow$) | **CLIP Sim** ($\uparrow$) |
| 1 | 15.93 | 0.631 |
| 2 | 15.32 | 0.629 |
| 4 | 15.32 | **0.642** |
| 8 | **15.31** | 0.641 |

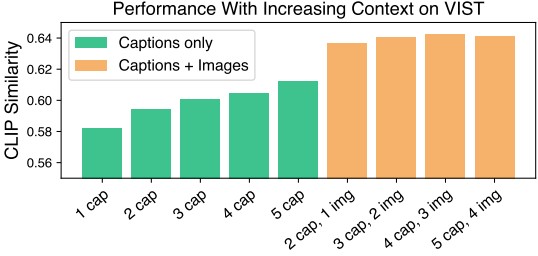

Figure 6: Performance of GILL on VIST generation.

Table 5: Contextual image retrieval on VIST (5 captions, 4 images as input). [†] indicates results from [31].

| | **VIST Recall@$k$** ($\uparrow$) | | |
|---|---|---|---|
| **Model** | **R@1** | **R@5** | **R@10** |
| CLIP ViT-L [43][†] | 8.8 | 22.3 | 29.8 |
| FROMAGe [31][†] | 18.2 | 42.7 | 51.8 |
| GILL (Ours) | **20.3** | **45.0** | **53.7** |

Similarity (0.622 to 0.645) and LPIPS (0.723 to 0.714). These results further highlight the efficacy of our model on handling long dialogue-like text inputs.

## 4.2 Qualitative Results

Finally, one of the more compelling applications of GILL is perhaps its ability to generalize to many different tasks, due to the LLM pretraining and freezing. We showcase several of these capabilities in Fig. 5. In many examples, we observed that GILL is able to outperform retrieval models such as FROMAGe [31] on examples where FROMAGe is unable to retrieve relevant images. GILL is also generally more sensitive to input context compared to Stable Diffusion [49], and can condition on *image* inputs, in addition to text, to generate more visually and semantically relevant image outputs.

## 5 Analysis

**Contextual Image Retrieval** In addition to generation, GILL is capable of image retrieval conditioned on image-text inputs. We run GILL on the VIST retrieval evaluation from [31]. We find that GILL performs comparably or better compared to prior approaches (Tab. 5). This shows that that the image generation objective does not cause image retrieval performance to deteriorate.

**The Effect of Context** GILL leverages an LLM backbone, which allows it to inherit some of the LLM's capabilities, such as improved sensitivity to long inputs. Fig. 6 shows that the performance of GILL generally improves with increasing input contexts on VIST [28]. In particular, when 2 captions and 1 image are provided as context, the model significantly outperforms the model with 5 text-only captions, highlighting the value of multimodal context over unimodal context.

**Generation-Only Objective** We investigate the effect of removing the retrieval loss (Eq. 4) from the training objective. On VIST (5 captions, 4 images), this ablated model achieves CLIP similarity of 0.636 and LPIPS of 0.694, which are comparable to scores of the original model (0.641 and 0.693 respectively). This suggests that the retrieval loss is not necessary for strong performance, although such a model would only be able to generate images and text and not retrieve images. These results also suggest that GILL is not bottlenecked by including the retrieval objective, and that it has sufficient capacity to perform both generation and retrieval.

**GILLMapper Module**  As described in Sec. 3.2, we propose the GILLMapper module, a lightweight transformer model that conditions on `[IMG]` embeddings and $q$ learnt embedding vectors. The output maps the LM embeddings into the input space of a text-to-image generation model, enabling image synthesis. We run several baselines to compare effectiveness, comparing our proposed model against (1) a linear layer, (2) a multilayer perceptron (MLP) with LeakyReLU activations, and (3) a 4-layer bidirectional transformer encoder. All models are conditioned on the $r$ `[IMG]` token embeddings from the LLM. Our results are presented in Tab. 3. GILLMapper is substantially better than these baseline models at learning the mapping from the frozen LLM to the Stable Diffusion generation model, as measured by Fréchet Inception Distance (FID) [25] on the CC3M validation set (which is a measure of image realism), and CLIP Similarity on VIST. On the VIST evaluation (which is out of distribution from CC3M), the other baselines perform significantly worse than GILLMapper, suggesting that they cannot generalize to longer sequences containing multiple images and texts.

**Number of `[IMG]` Tokens**  We experiment with varying the number of `[IMG]` tokens, $r$ (Tab. 4). As $r$ increases, generation generally improves, plateauing around $r = 4$. We observe that lower values of $r$ appear to result in worse results, as the inputs to GILLMapper are shorter and less expressive.

## 6  Conclusion

We proposed a method of mapping text-only LLMs to strong visual models. This enables them to learn to process arbitrarily interleaved image-and-text inputs, and output generated text, retrieved images, and generated images. We show that it is possible to efficiently learn a mapping between the embeddings of a frozen pretrained LLM and a frozen pretrained image generation model, and that doing so effectively boosts image generation for tasks that require stronger language context dependence. Finally, we also showcased several compelling qualitative results on a variety of multimodal tasks. Our approach is modular, and can benefit from stronger LLMs or visual models released in the future. Scaling up the LLM backbone, image generation backbone, or visual processing model, are promising directions that will likely induce even stronger vision-and-language capabilities.

## Acknowledgements

This work was partially supported by a gift from Cisco Systems, and by ONR N000142312368 and DARPA/AFRL FA87502321015. We thank Wendy Kua for help with the figures. We thank Jared Fernandez, Yutong He, Saujas Vaduguru, Yonatan Bisk, and our anonymous reviewers for feedback and helpful discussions on previous versions of this paper.

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

# A    Limitations

GILL relies on an LLM backbone for many of its capabilities. As such, it also inherits many of the limitations that are typical of LLMs. One limitation is the potential for hallucinations [6], where the model generates content that is false or not relevant to the input data. Another limitation of the model in generating text is in repetitions and neural text degeneration [27], where the model generates the same content multiple times. We also observed that the OPT-6.7B model also does not always consistently generate coherent dialogue text.

These limitations may be addressed by techniques that address hallucinations and degenerations in text-only LLMs, or by using improved LLMs that are less prone to these issues. In GILL, we used a 6.7B model. In the future, it will be valuable to scale up the approach with even larger LMs, or those trained with improved objectives [54], instruction finetuning [58] or human feedback [40]. Depending on downstream applications, using models trained explicitly on dialogue data [15] may also be helpful for dialogue capabilities (e.g., deploying multimodal chatbots).

With regards to the visual models, another limitation of our approach is in its limited visual processing. At the moment, we use only $k = 4$ visual vectors to represent each input image (due to computational constraints), which may not capture all the relevant visual information needed for downstream tasks. These vectors are produced by a frozen pre-trained visual encoder, and so the visual information in the vectors is heavily constrained by the pre-training task. As a result, the model may not always process images correctly or in enough detail to produce accurate or high-quality results. However, this limitation can potentially be addressed in the future by scaling up the visual model, using models with varied pre-training objectives that encode more visual information while still being mappable to the hidden space of the LLM, or using more sophisticated visual mappings [4, 33] that can capture a richer set of visual features. Similarly, we observed during inference that our model sometimes does not generate relevant images for certain types of prompts. We attribute this to our finetuning dataset being CC3M, which is relatively small compared to modern large scale image-text datasets [51]. It is likely that training GILLMapper on an even larger corpus of text data will improve its alignment to the image generation backbone.

One of the advantages of our model is that it is modular, and can benefit from stronger visual and language models released in the future. It is likely that it will also benefit from stronger text-to-image generation backbones, or through finetuning the generation backbone rather than just the GILLMapper module. We leave such scaling explorations for future work.

# B    Broader Impact

**AI Assistants**    Recent advances in dialogue based chatbots have sparked interest in using LLMs for interactive conversational applications. GILL is a multimodal language model capable of processing image and text inputs, and producing image and text outputs. These capabilities may enable a wider range of applications. For example, AI assistants which can produce image and text outputs would be able to answer a wider range of queries, providing visual content when necessary to illustrate certain points. Concrete applications may include creative endeavors (e.g., iteratively refining a generated image with instructions), answering questions that benefit from visual outputs (e.g., describing food items), and more. Scaling GILL and refining it with methods such as reinforcement learning from human feedback (RLHF) [32] are promising directions to improve the capabilities of multimodal AI assistant systems.

**Disinformation and Harms**    Aside from the technical limitations detailed in Sec. A, there are broader societal issues that should be considered with the development of generative models of text and images. LLMs have the potential to generate plausible sounding (but false) text [22, 6], propagating disinformation at scale. As GILL uses an LLM backbone, it is also susceptible to these potential issues. Furthermore, as multimodal generative models which can also produce image content, models such as GILL also introduce potential issues with producing even more convincing disinformation through interleaving text with realistic generated images. As GILL makes use of an image generation backbone, it is also susceptible to the risks that typical text-to-image generation models introduce, such as generating false images of real people. These harms may possibly be mitigated by introducing watermarking into generated images [37, 70], or by deploying systems to detect generated images [12].

**Bias and Safety**    GILL makes use of pretrained LLMs and multimodal models (such as CLIP [43] and Stable Diffusion [49]), which are trained on large, noisy, Internet-scraped data (such as LAION-400M [51]). Due to their curation process, these datasets often contain undesired biases, malignant stereotypes (see [8] for a comprehensive discussion on large scaled multimodal datasets). One advantage of GILL is that it is efficient to train and completely *modular*, allowing its components (i.e., the LLM, visual encoder, or image generator) to be swapped out for other pretrained models (for example, models which have been further calibrated to reduce unintended biases).

**Intended Uses**    GILL is a research prototype which showcases possible capabilities of multimodal language models which can both process and produce image and text outputs. Due to the limitations described above, GILL is not in its current state intended for deployment in practical applications, especially in high risk or sensitive domains without further analysis. At its current model scale (a 6.7B parameter LLM), GILL also lacks many of the abilities of larger language models [9], and applications would likely benefit from increased scaling of the LLM and visual models.

## C    Deciding to Generate or Retrieve

As detailed in Sec. 3.3, we evaluate several models on the annotated PartiPrompts [65] dataset. Each prompt is annotated with one of two labels: "ret" or "gen", indicating whether image retrieval or image generation produces a more appropriate image for the corresponding prompt. For example, the prompt *"a portrait of a statue of the Egyptian god Anubis wearing aviator goggles, white t-shirt and leather jacket, flying over the city of Mars."* is labeled as "gen", as there are (understandably) no appropriate images in the CC3M retrieval set, and generation produces a more relevant output. In contrast, *"the geyser Old Faithful"* is labeled as "ret," as there are very relevant candidate images available for this prompt. We evaluate several models for making this decision on the validation set (Tab. 6), evaluating using F1 score given the class imbalance of the dataset (201 "gen", 110 "ret" in the validation set labels):

1. **Baselines:** We measure the F1 score of several baseline methods, which provide a lower bound for how well data-driven approaches can do. We find that always retrieving an image, always generating an image, or simply deciding randomly (with a prior proportional to class frequencies) achieve F1 scores of 0.267, 0.389, and 0.451 respectively.

2. **Heuristic:** We also consider a simple heuristic which considers the maximum cosine similarity of the retrieval embedding against the entire image candidate set (i.e., the training set of CC3M). We run a grid search from 0 to 1 for possible threshold values. Whenever the maximum cosine similarity is above a threshold, we return "ret" and "gen" otherwise. This achieves an F1 of 0.261 – 0.559, depending on the threshold used (a threshold of 0.5 gives F1 of 0.261).

3. **Linear classifier:** Lastly, we train a linear classifier that takes as input the outputs of the LLM for the `[IMG]` tokens and the maximum cosine similarity. This classifier is trained with the binary cross-entropy loss over the training set of PartiPrompts annotations. This linear classifier achieves an F1 score of between 0.393 – 0.552, depending on the probability threshold used (a threshold of 0.5 gives an F1 score of 0.547).

We use the linear classifier in our final model, as it requires less hyperparameter tuning compared to the heuristic baseline, and performs comparably on quantitative metrics. During generation of qualitative samples (Fig. 7 and Fig. 5 in the main paper), we observed that the linear classifier generally performed well for many prompts, and decided correctly whether to retrieve or generate.

## D    Qualitative Results

We present further qualitative samples in Fig. 7. We find that GILL is able to process complex text prompts more effectively than Stable Diffusion for many examples in PartiPrompts [65]. On VisDial [16] dialogue inputs, GILL is able to generate more relevant outputs (as measured against groundtruth images). We attribute these improved results to the stronger text representations of the LLM, and the effectiveness of our GILLMapper network.

Table 6: Results on PartiPrompts for classifying retrieval or generation.

| Method | F1 |
|---|---|
| Always retrieve | 0.267 |
| Always generate | 0.389 |
| Random | 0.451 |
| Heuristic | 0.261 − 0.559 |
| Linear classifier | 0.393 − 0.552 |
| Human performance | 0.851 |

Table 7: Results on image captioning on MS-COCO (2017) [34] and VQA [24]. For captioning, we report BLEU [41] and METEOR [5] scores. For VQA, we report the accuracy after applying the normalization described in the VQA repo (`https://github.com/GT-Vision-Lab/VQA`). † indicates our reimplementation.

| Model | Captioning | | VQA |
| | BLEU-4 | METEOR | 0-shot |
|---|---|---|---|
| Frozen† [56] | - | - | 0.2553 |
| MAGMA [19] | - | - | 0.2835 |
| FROMAGe [31] | 0.1023 | 0.2873 | 0.2851 |
| Ours | 0.1059 | 0.2529 | 0.3178 |

# E  Other Evaluations

## E.1  Comparison to Prior Multimodal LMs

We ran evaluations on VQAv2 [24] and image captioning on MS-COCO [34]. The results are presented in Tab. 7. We found that GILL is comparable to models trained with similar compute and data. On VQAv2, we achieve a zero-shot val accuracy of 0.3178, which is slightly better than prior approaches of similar model sizes and compute: FROMAGe [31] achieves a zero-shot accuracy of 0.2851, Frozen [56] achieves 0.2553, and MAGMA [19] achieves 0.2835. For image captioning on the MS-COCO (2017) validation set, GILL achieves a BLEU@4 of 0.1059 and METEOR of 0.2529, which is comparable to FROMAGe (BLEU@4 of 0.1023 and METEOR of 0.2873). GILL is also capable of a wider set of tasks (e.g., generating interleaved image and text outputs) compared to these models.

We note that these scores are lower than SOTA models, as they are usually much larger and trained with significantly more compute and data (e.g., Flamingo [4] uses 23,040 TPU days, BLIP-2 [33] uses 144 GPU days, while ours uses 4 GPU days). Scaling up GILL to similar data and parameter scales to further push its capabilities is an exciting avenue for future work.

## E.2  Increasing Context on VisDial

GILL leverages an LLM backbone, which allows it to inherit some of the LLM's capabilities, such as improved sensitivity to long input contexts. In the main paper, we showed that GILL can better condition on longer image and text inputs to generate more relevant images for VIST [28]. We run a similar experiment on Visual Dialogue [16], varying the number of dialogue rounds provided as input context to GILL and Stable Diffusion (SD) [49].

The results are presented in Fig. 8. We find that when longer text context is provided to both models, the performance of generating relevant images steadily improves. Interestingly, SD performance plateaus after 6 rounds of dialogue, while GILL continues to improve, outperforming SD when 7 or more rounds of dialogue are provided. These results showcase the improved sensitivity of our model to conditioning on long, dialogue-like text. Despite both approaches using the same image generation backbone, GILL is able to better make use of longer dialogue-text inputs (despite being only finetuned on image caption data).

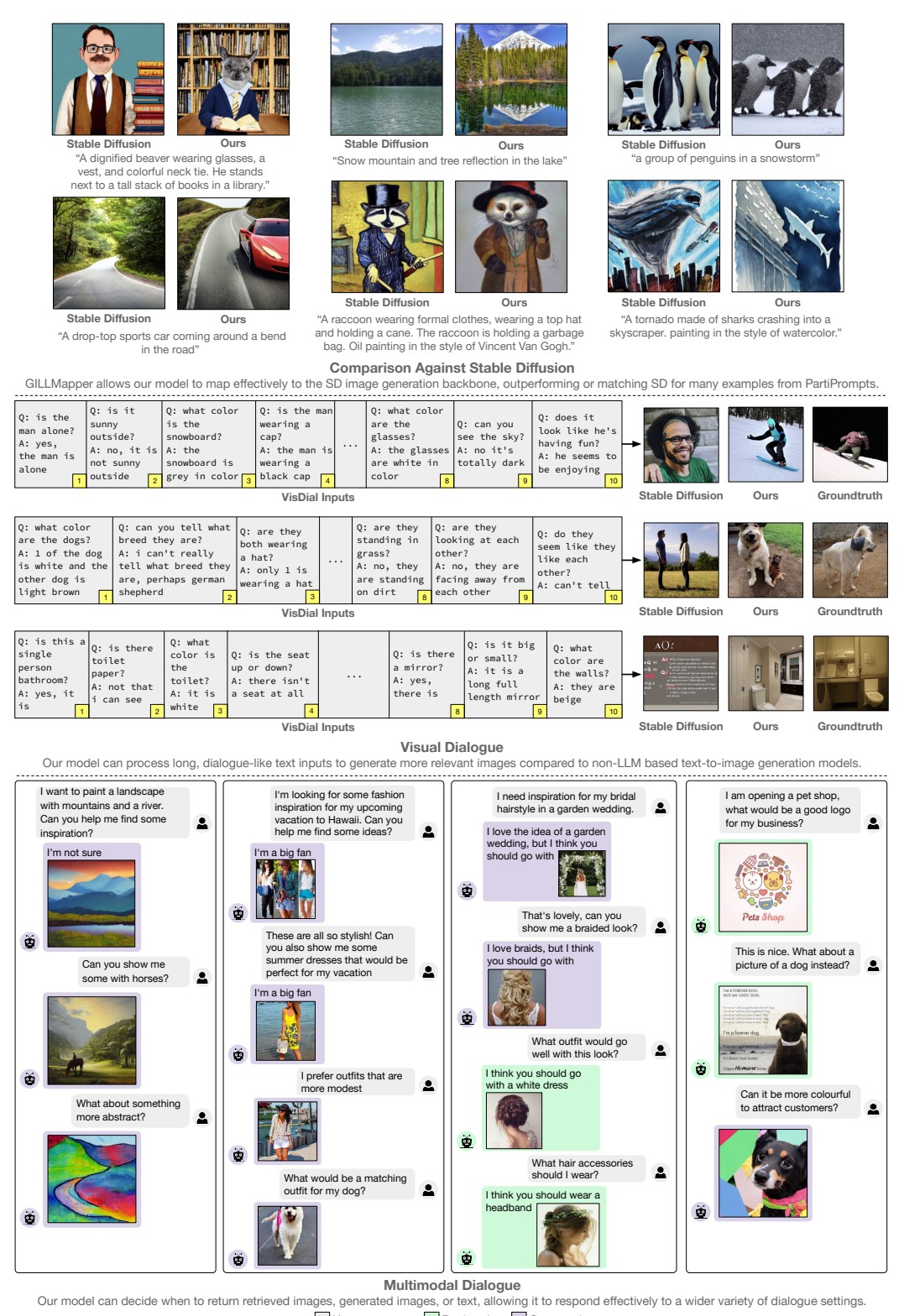

Figure 7: Further qualitative samples from GILL. It is more sensitive to text inputs due to its LLM backbone, and better at processing complex text prompts.

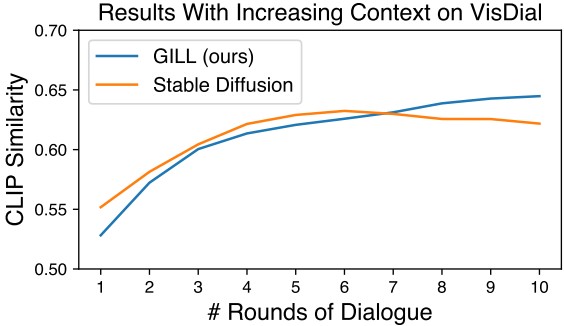

Figure 8: Performance of our model and Stable Diffusion [49] with increasing context for generating VisDial [16] images. Our model is able to better process long dialogue-like text descriptions.

Table 8: Zero-shot FID [25] on the MS-COCO [34] (2014) validation set. 30,000 random samples are used to evaluate all models.

| Model | FID ($\downarrow$) |
|---|---|
| GLIDE [38] | 12.24 |
| Make-A-Scene [21] | 11.84 |
| DALL-E 2 [44] | 10.39 |
| LAFITE2 [72] | 8.42 |
| Imagen [50] | 7.27 |
| Parti [65] | 7.23 |
| Re-Imagen [14] | 6.88 |
| SD [49] v1.5 | 9.22 |
| GILL (ours) | 12.2 |

### E.3 Image Generation

In addition to our evaluations on VIST [28] and VisDial [16], we also run evaluations on our model's ability to generate images from MS-COCO [34] captions (Tab. 8). We generate images using 30,000 randomly sampled captions from the MS-COCO (2014) validation set, which is the standard evaluation of text-to-image generation models. We report zero-shot FID scores [25] of our model, Stable Diffusion [49] v1.5 (which we use as our backbone image generator), and several other approaches in Tab. 8. For our generation results and SD results, we use a classifier-free guidance scaling factor of 3.0 and 250 DDIM inference steps. On MS-COCO, our approach achieves a worse FID score than SD (9.22 to 12.2). This is likely because this task does not benefit as much from the LLM backbone, which has not been trained on as many image captions as SD (which exclusively trains on caption-like data). These numbers will likely improve further by finetuning GILL on even more text data (including image captions), which will allow our model to align more closely to the input space of the SD image generator.

### E.4 Inference Speed

One concern for deploying LLMs is the inference throughput and speed. We benchmark the inference performance of GILL on a single A6000 GPU. Generating text has the same throughput as a regular LM of the same size (i.e., that of OPT 6.7B). The main increase in inference time occurs when the model produces `[IMG]` tokens. For a batch size of 1, if the model decides to retrieve images, the additional inference overhead is minimal (less than 0.001s on average) as image retrieval is fast, requiring just a single matrix multiplication followed by a max operation. However, if GILL generates an `[IMG]` token, it takes 3.5s on average for the Stable Diffusion backbone to generate a corresponding image.

Overall, GILL's inference speed is bottlenecked by the frequency of image generation, which is dependent on the application domain. In the case of generating dialogue-like text, we observed that images are usually generated or retrieved once or twice in a natural conversation. Amortized over a long conversation, it does not lead to a significant increase compared to a text-only LLM, though exact numbers would depend on the application.

## F   Human Annotation on PartiPrompts

In Sec. 3.3 of the main paper, we described the process of annotating PartiPrompts [65] with per-example labels to retrieve or generate. The interface shown to human annotators is shown in Fig. 9. Annotators are tasked to determine which of two anonymized images are (1) more relevant to the provided prompt, and (2) more realistic. We randomize the order of the two images as well (i.e., the output of the retrieval model shows up 50% of the time as Image A).

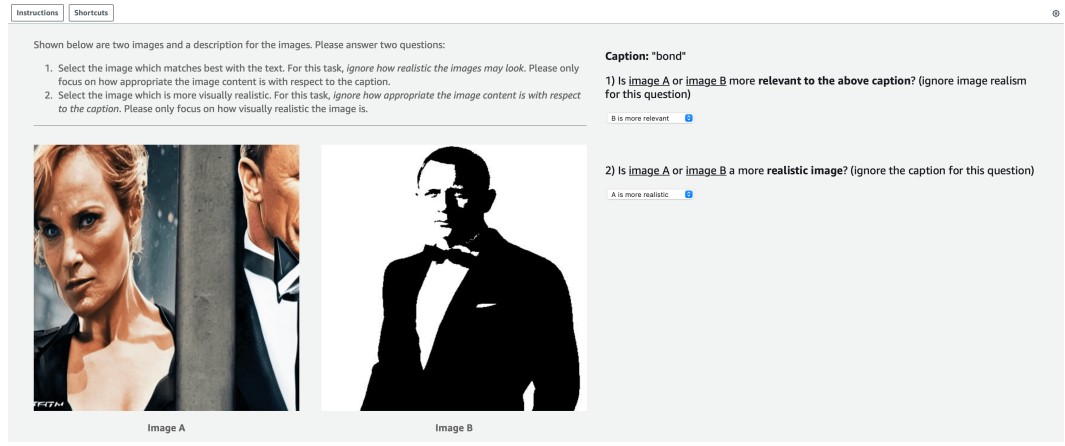

Figure 9: User interface shown to human annotators for annotating PartiPrompts [65] examples.

We show each example to 5 independent human annotators. For determining whether to label a particular example as "ret" or "gen", we take the majority vote of the 5 annotators on the image relevance question ("Is image A or image B more relevant to the above caption?"), and only keep the examples with an inter-annotator agreement of at least 4/5. This results in approximately 900 examples remaining (out of the 1,632 examples in PartiPrompts). Our annotations will be publicly released to facilitate future evaluations on this task.

We conducted evaluations on the Amazon Mechanical Turk platform with human annotators located in the US and Canada. Annotators were paid at an estimated hourly rate of 15 USD per hour. In total, we spent approximately 326 USD to collect these annotations.

