# Appendix:
# Generating Images with Multimodal Language Models

## Abstract

We detail current limitations of GILL, and suggest possible directions to alleviate
this in future work. We also describe the broader impact of our work, including
possible applications, risks, and intended uses. Finally, we provide more quantita-
tive and qualitative evaluations, including results on deciding whether to retrieve
or generate, results on the effect of increasing context on VisDial, text-to-image
generation results on MS-COCO, and present more qualitative samples from GILL.