# OpenReview forum: "Generating Images with Multimodal Language Models"
_NeurIPS.cc/2023/Conference — NeurIPS 2023 poster_

### Official Review · Reviewer_Gk8n · 2023-07-06

**Soundness:** 3 good
**Presentation:** 2 fair
**Contribution:** 4 excellent
**Rating:** 6
**Confidence:** 4

**Summary:**

The paper introduces a new method called GILL that effectively integrates frozen LLMs with pre-trained image encoder and decoder models to create coherent image and text outputs. The authors show GILL's superior performance over baseline generation models in tasks involving longer and more complex text and its capability for image retrieval and generation at inference. GILL extends the capabilities of pre-trained LLMs to multimodal by mapping the embedding spaces of the LLMs to visual models with a mapping network. The proposed framework is inspiring and interesting. It may have a large impact for unified multimodal pre-training.

**Strengths:**

* The proposed approach is innovative and inspiring, combining frozen LLMs with visual models to generate coherent image and text outputs. This approach is efficient, as it doesn't require training the image generation model from scratch.
* The paper is well-written, logically structured, and accessible.
* Experimental results show that GILL is effective in processing long-form text and generating images that are more closely matched to the text than baselines. It can process arbitrarily interleaved image-text inputs and generate retrieved images, novel images, and text, which expands its capabilities compared to previous models.
* The paper’s approach to aligning visual tokens with LLMs is promising, and the proposed solutions are insightful.

**Weaknesses:**

* The paper lacks clear implementation details. Despite Eq. 5 explaining the overall objective to optimize, the parameters to be updated are not clearly stated in the equations. This makes it unclear which parts of the model require training and which parts remain fixed.
* While the GILL framework is novel, the GILLMapper idea is not new. Aligning a pre-trained/new encoder with the CLIPText encoder of the Stable Diffusion model has been previously explored [1,2], and this related literature is not discussed in the paper.  [1] discussed altering the Language Encoder to extend the multilingual capacities for Stable Diffusion. [2] plug-ins the multimodal encoders to replace the CLIPText of the Stable Diffusion. The Eq. 3 of GILL paper is similar to the Eq. 3 of [2] even though GILL applies visual tokens here.

[1] AltCLIP: Altering the Language Encoder in CLIP for Extended Language Capabilities. arXiv preprint arXiv:2211.06679 (2022).

[2] GlueGen: Plug and Play Multi-modal Encoders for X-to-image Generation. arXiv preprint arXiv:2303.10056 (2023)

* The paper lacks comparison with state-of-the-art multimodal methods such as BLIP.
* The paper doesn’t discuss other relevant works, such as one [3] that unifies both text-to-image and image-to-text in one framework.

[3] CoBIT: A Contrastive Bi-directional Image-Text Generation Model." arXiv preprint arXiv:2303.13455 (2023).

* The paper lacks discussion of the inference speed, which is an especially essential factor for diffusion-based image generation task.
* The paper does not address the possibility of the OPT model not perfectly aligning with CLIPText, which could create cross-model gaps  that are challenging to bridge. It's unclear how the authors would tackle such challenges and whether these difficult-to-address domain gaps would cause a decrease in performance.

**Questions:**

See weakness above

**Limitations:**

See weakness above

---

> ### Author Rebuttal · Authors · 2023-08-07
>
> We thank the reviewer for their valuable comments. We are glad that the reviewer found our proposed approach innovative and inspiring, and recognized that it is efficient. We are pleased that the reviewer found the paper well-written, and appreciated GILL’s improved capabilities over baseline models.
>
> ## 1. Implementation Details
> > Despite Eq. 5 explaining the overall objective to optimize, the parameters to be updated are not clearly stated in the equations
>
> We describe the trainable parameters in L178-180, which are the linear layers $\\mathbf{W}\_{\\text{i2t}}$, $\\mathbf{W}\_{\\text{t2i}}$, $\\mathbf{W}\_{\\text{cap}}$, the IMG embedding matrix $\\mathbf{E}\_{\\text{img}}$, and the GILLMapper parameters $\omega$ and query vectors $q\_{1:L}$ (while everything else, including the LLM backbone, the visual encoder, and the SD generator, remain frozen), which is also illustrated in Fig. 2. We will also update Eq. 5 to read:
>
> $$\\min\_{\\mathbf{W}\_{\\text{i2t}}, \\mathbf{W}\_{\\text{t2i}}, \\mathbf{W}\_{\\text{cap}}, \\mathbf{E}\_{\\text{img}}, \\omega, q\_{1:L}}
> \\frac{1}{N} \\sum\_{i=1}^N \\big(l\_c(\\mathbf{x}\_i, \\mathbf{y}\_i) +  l\_p(\\mathbf{y}\_i) +  l\_g(\\mathbf{y}\_i) +   l\_{r}(\\mathbf{x}\_i, \\mathbf{y}\_i) \\big) $$
>
> which hopefully clarifies. We will also release **pretrained models weights and the full code** for training a model from scratch, to account for any other remaining code and low-level implementation details that are difficult to explain in the space limitations of the paper.
>
> ## 2. Discussion of prior work
>
> > While the GILL framework is novel, the GILLMapper idea is not new …
>
> > The paper lacks comparison with state-of-the-art multimodal methods such as BLIP.
>
> Thanks for the pointers, and we are glad that the reviewer recognized the novelty of the overall GILL framework. We will update the paper to include these references in our discussion on related work with respect to the GILLMapper module.
>
> In this paper, we primarily focus on the ability of GILL to generate images conditioned on interleaved image-text inputs, since most existing multimodal models such as BLIP cannot do this. For these reasons, we primarily compared against Stable Diffusion to test their image generation abilities. We also compared retrieval performance against FROMAGe, which is one of the few prior approaches which can process text + image inputs to generate text + image outputs (although FROMAGe retrieves rather than generates). We showed that GILL is better than SD at generating images with longer contexts (Table 1, 2) and as good as FROMAGe in retrieving images (Table 5). We will also include results on VQAv2 and MS-COCO captioning (see response to reviewer QYwk), which show that GILL is competitive with models of similar sizes and amount of training on image-to-text generation.
>
> > The paper doesn’t discuss other relevant works, such as one [3] that unifies both text-to-image and image-to-text in one framework.
>
> CoBIT is concurrent work according to NeurIPS guidelines (< 2 months of the deadline), but we will update the paper to briefly discuss it. To the best of our understanding, CoBIT is capable of generating text *or* image, but does not generate *interleaved* text and image outputs. It does not include methods to determine when to synthesize text vs. images. Our work is capable of generating text and images (by representing images as `[IMG]` tokens). This allows us to generate multimodal dialogue-like outputs consisting of both images and text.
>
> GILL is a proof of concept for multimodal models that can process image + text and produce image + text, trained in an efficient manner (4 GPU days). Scaling it up to the data/compute scale of BLIP-2 (144 GPU days) and CoBIT (6144 TPU days) is a promising direction for future work.
>
> ## 3. Inference Speed
>
> Generating text has the same throughput as a regular LM of the same size (i.e., that of OPT 6.7B). The main increase in inference time occurs when the model produces `[IMG]` tokens. For a batch size of 1, if the model decides to retrieve images, the additional inference overhead is minimal (< 0.001s on average) as image retrieval is fast (requiring a single matrix multiplication followed by a max operation). If the model predicts to generate an image, it takes 3.5s per image on average on a single A6000 GPU, which is the time for SD to generate a single image.
>
> Overall, GILL’s inference speed is bottlenecked by the frequency of image generation, which is dependent on the application domain. In the case of generating dialogue-like text, we observed that images are usually generated or retrieved once or twice in a natural conversation. Amortized over a long conversation, we believe it does not lead to a significant increase compared to a text-only LLM, though exact numbers would depend on the application.
>
> ## 4. Alignment
>
> Our experiments on VIST and VisDial (out of domain w.r.t. the CC3M finetuning data) suggest that our current approach does well at aligning the `[IMG]` representations with the CLIP text encoder. With respect to domain shift, our approach does not simply overfit to CC3M: our model surpasses SD on VIST and VisDial. With respect to the modality gap, both SD and our approach use the same image generation backbone, and the only difference is in the text encoder. If the learnt mapping was poor, we would not be able to outperform SD on these benchmarks. These results suggest that the current approach of training the GILLMapper module aligns it well to the CLIP text space.
>
> However, we agree that it is possible that on more difficult or more out of distribution tasks, the `[IMG]` representations may not be mapped well. A possible method to alleviate this would be to include the SD generator into the pipeline, and train the whole model end-to-end with the image generation loss rather than the L2 loss on the text encoding. This would be useful to explore in future work, but would likely require significantly more GPU memory.

---

> > ### Comment · Reviewer_Gk8n · 2023-08-14
> > **After Rebuttal**
> >
> > Thanks to the detailed explanation from authors. Most of my concerns are addressed and I'd like to raise the score to WA.

---

> > > ### Author Response · Authors · 2023-08-14
> > >
> > > We are happy to hear that we've addressed most of the concerns, and we thank the reviewer again for the feedback! Please let us know if there are any further clarifications we can provide.

---

### Official Review · Reviewer_QYwk · 2023-07-07

**Soundness:** 3 good
**Presentation:** 2 fair
**Contribution:** 3 good
**Rating:** 7
**Confidence:** 4

**Summary:**

The paper proposes a novel method to stitch together LLMs and text conditional image diffusion models, to process interleaved image-text and output interleaved image-text. The paper evaluates their methodological contributions on several tasks where outputting images is required.

**Strengths:**

S1. The ability to output generated image or text OR retrieve images to satisfy a user’s query is not something I have seen in prior work and speaks to the generality of the formulation.

S2. The paper explores how to stitch together large pre-trained models (i.e., LLMs and image diffusion models). It represents a creative exploration of stitching these models together using a combination of learnable modules, captioning losses, contrastive losses, and distillation losses, which is non-trivial exploration.

S3. Generative applications of the model appear impressive in the demo figures.

S4. Ablations related to image generation and retrieval are generally strong, thoughtful, and cover many different key design decisions.

S5. Fine-tuning cost is cheap and reasonably accessible (~96 A6000 GPU hours).


**Weaknesses:**

W1. Missing citation. The following paper conducts probes showing that pre-trained image and text representations can be aligned: [1] Ilharco et al. Probing Contextual Language Models for Common Ground with Visual Representations. 2021.

W2. Missing citations. Since one potential use-case of the model is image captioning (i.e., the model can handle image input and textual output), I also suggest citing recent captioning work: [2] Yu et al. CoCa: Contrastive Captioners are Image-Text Foundation Models. 2022.; [3] Li et al. BLIP: Bootstrapping Language-Image Pre-training for Unified Vision-Language Understanding and Generation. 2022.; [4] Li et al. BLIP-2: Bootstrapping Language-Image Pre-training with Frozen Image Encoders and Large Language Models. 2023; [5] Dai et al. InstructBLIP: Towards General-purpose Vision-Language Models with Instruction Tuning. 2023.

W3. Missing citation, unsupported claim. Recently, unified models take image and text inputs and generate image and text outputs for a variety of tasks. e.g., [6] Lu et al. Unified-IO: A Unified Model for Vision, Language, and Multi-Modal Tasks. 2022. Hence, lines 4-6 in the abstract seem a bit overstated: “Ours is the first approach capable of conditioning on arbitrarily interleaved image and text inputs to generate coherent image (and text) outputs.”

W4. Presentation. Consider adding percentage point improvements over baselines in the abstract and intro to give the reader an idea early on of what kinds of gains can be expected by using your model over competitors.

W5. Presentation. Consider giving some more intuition about why you are making certain design decisions, instead of just saying what you are doing. For example, I found the motivation for your introduced IMG1, IMG2, …, IMGr tokens to be a bit confusing. It seems here, the supervision encourages learning when to produce IMG tokens, but not what content should be captured in the representation space, which would be different depending on the image. Why should this design decision will lead to learning good last layer hidden representations for arbitrary images.

W6. Clarity. The Image retrieval section seems incomplete. An objective is provided, but it is not clear to me how the model is ultimately used to retrieve images at evaluation time.

W7. Clarity. The work proposes to use a learned mapping module (GILLMapper) to go from text hidden states (of an LLM) to vision model embedding space (of a text-conditioned image diffusion model). However, it is not clear this module is needed. An alternative strategy might be to feed generated LLM text directly to the text-conditioned diffusion model, thereby bypassing the need to map representations.

W8. Missing evaluation, unsupported claims. The paper claims that the model can generate image or text or retrive images. However, the experiments test only the abilities related to outputting images. How does the model perform on tasks that require outputting text (e.g., CoCo CIDEr and VQAv2 accuracy)? I consider such evals to be critical to verify claims that the model can generate sensible text outputs based on image inputs. Such evaluations also allow for comparison with Flamingo-like models.

W9. Generality of the method. Deciding whether to retrieve or generate images depends on a liner classifier that is dataset specific. This means that a practitioner wanting to use this model out-of-the-box on data of their own may have to train their own classifier (annotate data etc.) to use the model.

W10. Clarity. How is CC3M turned into an interleaved training dataset for model training? Are there any data sampling strategies that are important here? How many images and captions are sampled? Are non-interleaved sequences also trained on (i.e., image or text only)?


**Questions:**

Here are my main questions distilled from the weaknesses above:

Q1. Can some of the clarity related questions be addressed, specifically W5, W6, W7, W10?

Q2. How does your model perform on CoCo captioning and VQAv2 (W8)? These also seem like useful evaluations to make sure that your model interprets the content in images.

Q3. Can authors address the concerns related to the generality of the method for image generation vs. retrieval (W9)?

**Limitations:**

No limitations or failure analysis is presented in the main paper. I suggest discussing conditions under which the model fails.

---

> ### Author Rebuttal · Authors · 2023-08-07
>
> We thank the reviewer for their valuable comments. We are glad that the reviewer recognized the creativity and generality of our approach in combining pretrained LLMs and visual models, and appreciated the impressive qualitative results, strong ablations, and accessible finetuning cost. We address specific queries below, and will incorporate all feedback.
>
> ## 1. Clarity Related Queries
>
> ### W5
> Our approach includes losses to supervise both learning when to produce `[IMG]` tokens, but also the representation of the `[IMG]` tokens (detailed in L120-140 of the main paper) through minimizing the L2 loss against CLIP text embeddings. The representations depend on context, so different text inputs produce different `[IMG]` token hidden states (due to the LM attending to different things). This allows us to produce the appropriate representations for different text inputs. We will update Sec 3.2 and the caption of Fig. 2 to make this clear.
>
> ### W6
> During inference, we follow standard procedure [7] and retrieve the image with the highest cosine similarity (between image embeddings and `[IMG]` token embeddings) from a dataset. For VIST (Tab. 5), this is from its val set (consistent with [8]). For other qualitative results, we retrieve from CC3M.
>
> ### W7
> One benefit of our approach is that it treats image outputs as continuous embeddings to be directly fed into an image generator. This allows us to leverage the capabilities of LLMs, such as in-context learning and the ability to handle longer inputs than can be handled by the SD text encoder (i.e., 77 tokens) for generating images. Continuous representations bypass the text bottleneck, and can be optimized completely end-to-end. Feeding generated text would require RL-like approaches, which may not be as straightforward. Please also see our response to reviewer bnn7, which shows that a text-only approach (using generated text from much larger GPT-3.5/4 models) underperforms GILL on VIST, suggesting that using generated text may be insufficient for more complicated tasks.
>
> ### W10
> We follow [8] in packing two random examples during training with probability 0.5. This means that 50% of the time, input data is `<img1><txt1><img2><txt2>`, while the other 50% of the time, it consists of single examples `<img1><txt1>`. Although the examples are distinct (`<img1>` is in general unrelated to `<img2>`), we find this helps encourage the model to attend to the correct image, rather than always attending to the first image.
>
> > Are non-interleaved sequences also trained on (i.e., image or text only)?
>
> We do not train on image or text-only sequences, though the frozen LLM was originally pretrained on text-only data.
>
> ## 2. VQAv2 and COCO
> We ran evaluations on VQAv2 and MS-COCO captioning, and found that GILL is comparable to models trained with similar compute and data. On VQAv2, we achieve a zero-shot val accuracy of 31.78, which is slightly better than prior approaches of similar model sizes and compute: FROMAGe achieves zero-shot accuracy of 28.51, Frozen achieves 25.53, and MAGMA achieves 28.35. On MS-COCO, GILL achieves a BLEU@4 of 0.1059 and METEOR of 0.2529, which is comparable to FROMAGe (BLEU@4 of 0.1023 and METEOR of 0.2873). GILL is also capable of a wider set of tasks (e.g., generating interleaved image and text outputs) compared to these models.
>
> We note that these scores are lower than SOTA models, as they are usually much larger and trained with significantly more compute and data (e.g., Flamingo uses 23,040 TPU days, BLIP-2 uses 144 GPU days, while ours takes 4 GPU days). Scaling up GILL to similar data and parameter scales to further push its capabilities is an exciting avenue for future work.
>
> ## 3. Generality of the Decision Method (W9)
> Our evaluations were conducted on PartiPrompts (P2), and we find that the decision classifier trained on P2 does reasonably well on a held out subset (Tab. 1 in the appendix).
>
> > This means that a practitioner … may have to train their own classifier
>
> We agree that this is a fair point, and whether it would generalize would likely depend heavily on the actual application’s data distribution, and the amount of data available. P2 is best suited to generalizing to domains that require separating factual from non-factual captions, which we believe is a key consideration for many downstream tasks. However, there is likely room for improvement in future work, e.g., by taking into account the quality of the generated image before making a decision.
>
> ## 4. Missing Citations
> ### W1 and W2
> We will update the intro to include [1]. We will also update the related work section to discuss image captioning approaches as suggested.
>
> ### W3
> > Recently, unified models take image and text inputs and generate image and text outputs for a variety of tasks. e.g., [6] …
>
> Thanks for the pointer! To the best of our knowledge, ours is still the first approach capable of generating coherent images and text. Unified-IO generates images *or* text, but without modifications, does not seem capable of producing outputs that are interleaved image and text, or decide when to generate text and when to generate images.
>
> In contrast, GILL generates text interleaved with images, and automatically determines when to produce images instead of text. CM3 and FROMAGe are capable of doing so, but CM3 frequently generates incoherent or low quality images (see appendix of [8]). We are able to do better as we leverage a strong pretrained diffusion model. FROMAGe retrieves images but does not generate novel images.
>
> ## 5. Limitations / Failure Analysis
> We included discussions on limitations, failure modes, potential future directions to alleviate these, and the broader impact of our model in Appendices A and B.
>
>
> ### References
> [7] Radford, Alec, et al. "Learning Transferable Visual Models From Natural Language Supervision." ICML, 2021.
>
> [8] Koh, Jing Yu. et al. “Grounding Language Models to Images for Multimodal Inputs and Outputs”. ICML, 2023.

---

> > ### Comment · Reviewer_QYwk · 2023-08-14
> >
> > Thanks to the authors for their extensive rebuttal. I generally feel more positively about the paper, especially with the newly reported numbers on VQAv2 and CoCo (thanks for running this!). I think it is important that the authors add these results to the main paper. I still think that the domain specific classifier is a fundamental weakness, but do not feel this is sufficient grounds for rejection. I am happy to raise my score to a 7.

---

> > > ### Author Response · Authors · 2023-08-14
> > >
> > > We thank the reviewer for their detailed feedback, and are happy to hear that our rebuttal addressed their concerns! We will definitely add these text generation results to the next version of the paper, and add some discussion about the domain specific classifier.
> > >
> > > Please let us know if there are any further clarifications we can provide.

---

### Official Review · Reviewer_5sAf · 2023-07-07

**Soundness:** 4 excellent
**Presentation:** 3 good
**Contribution:** 4 excellent
**Rating:** 8
**Confidence:** 4

**Summary:**

The authors train adapters to map embeddings of pre-trained image encoders and decoders to pre-trained LLMs. This allows them to input interleaved images with text into a pre-trained LLM and also make the LLM generate [IMG] tokens as required, which can be fed into a decoder to generate images or can be used to retrieve from a set of images based on cosine similarity.

**Strengths:**

- Clean idea that introduces a fundamental novelty in the capability of text-to-image models - the ability to interleave image and text to generate images.
- Adapting image to text space to use with pre-trained LLM's is neat since we don't need to train yet another large model. This idea has been shown to work well in contemporary models like LLaVA and miniGPT-4, but they show it works well for generation as well - a capability these other contemporary models lack.
- The results look very promising. They show longer context helps, further justifying the need of such models that can handle long chains of interleaved image and text.
- Such a line of work can have multiple interesting follow-up works on evaluating compositionality, etc.

**Weaknesses:**

- One of the strengths of interleaving image and text is that one can expect to extract different concepts from different images to compose a new image. For instance: "a cup that looks like <image_of_come_cup>, but on a table that looks like <image_of_some_table>" Such an analysis could have been great similar to the goal of this paper: https://arxiv.org/pdf/2212.04488.pdf.

- Some other recent methods aim to solve similar goals as advertised in the qualitative results. For instance, Figure 5 top row - papers like prompt-to-prompt and imagick aim to do this. Although their method cannot handle arbitrary interleaved images and text, a qualitative comparison of these applications that both methods can handle would make the paper very strong.

**Questions:**

Is it necessary to train for both generation and retrieval? How does it work if we just train for generation and not include the retrieval loss? The ablation in the supplemental Table 1 is just during inference, correct?

During training, the authors say they use CC 3M. This only has one image and text pair. Do they use any data that has multiple images and text? Are results on Visual Stories without training on it?

What are the [IMG_r] tokens learning exactly and how does varying `r` (the number of image tokens) affect results? I am guessing each of the tokens are learning some
 image concepts present frequently among the images in data? An analysis on this would have been great!

**Limitations:**

Authors discuss limitations in the supplemental.

---

> ### Author Rebuttal · Authors · 2023-08-07
>
> We thank the reviewer for their thoughtful and valuable comments. We are glad that the reviewer appreciated the promising results from our paper, the possibility for interesting follow-up work, and recognized that it is a clean idea which introduces fundamental capabilities (image generation) to multimodal language models. We address specific queries below, and will incorporate all feedback.
>
>
> ## 1. Training a generation-only model
> We ran the suggested experiment, removing the retrieval objective from the training losses. On the VIST evaluation (5 captions, 4 images), this ablated model achieves CLIP similarity of 0.636 and LPIPS of 0.694, which are comparable to that of the original model (CLIP similarity of 0.641 and LPIPS of 0.693). This suggests that it is not necessary to include the retrieval loss, although such a model would only be able to generate images and text and not retrieve images. These results also suggest that the model is not bottlenecked by including the retrieval objective, and that there is sufficient capacity for the model to perform both generation and retrieval.
>
>
> ## 2. Training data
>
> Our model is trained on just the CC3M dataset (which contains just single image-text pairs). We follow [1] in randomly packing two distinct examples together during training with probability 0.5. This means that 50% of the time, examples consist of `<image1><text1><image2><text2>`, while the other 50% of the time, it consists of single examples `<image1><text1>`). Although the two examples are distinct (`<image2>` is in general unrelated to `<image1>`), we also find this helps encourage the model to attend to the correct image in the sequence, rather than simply always attending to the first image in the input sequence.
>
> > Are results on Visual Stories without training on it?
>
> This is correct. Despite not using explicitly interleaved data with multiple related images (such as Flamingo [2] or CM3 [3]), GILL is capable of processing Visual Stories zero-shot (which consists of 5 captions, 4 images, more than even the 2 images + 2 captions we randomly introduce through packing). We attribute this to the ability of the pretrained frozen LLM to generalize to multiple images, as we learn to map the images to embeddings in the LM space.
>
>
> ## 3. Analysis on the $r$ image tokens
>
> In the main paper, we included ablations and discussion on how varying the number of tokens $r$ affects generation results (Table 4, and L275-277 on page 9). We find that lower values of $r$ (e.g., $r=1$ or $r=2$) tends to result in worse results on the VIST task, as the model is less expressive, which motivates our design choice of using $r=8$ tokens for improved generated image quality and image-text match (as measured on VIST).
>
> > I am guessing each of the tokens are learning some image concepts present frequently among the images in data
>
> We largely agree with this intuition: the $r$ image tokens are used as inputs to GILLMapper (trained by minimizing the L2 loss of its outputs against the CLIP text embeddings), whose outputs are used by the Stable Diffusion image generator. The “concepts” learnt by the image tokens in GILL would likely correspond to the “concepts” used by SD to produce image tokens. Further and more comprehensive analysis would likely be necessary to verify this (e.g., learning linear probes, or finding a way to visualize the embeddings of the individual tokens without significantly altering the SD pipeline), which we believe might be out of the scope of this paper, but would be very interesting to study in future work.
>
>
> ## 4. Extracting concepts from images
>
> > One of the strengths of interleaving image and text is that one can expect to extract different concepts from different images to compose a new image. For instance: "a cup that looks like <image_of_come_cup>, but on a table that looks like <image_of_some_table>" Such an analysis could have been great similar to the goal of this paper: https://arxiv.org/pdf/2212.04488.pdf.
>
> We agree that this is a very exciting direction for follow up work! One of the limitations of GILL (discussed further in Appendix A) is that it has somewhat limited visual processing capabilities. Hence, we don’t expect that it will do as well as the referenced paper or prompt-to-prompt on tasks that involve finegrained image editing.
>
> In GILL, input images are represented as $k = 4$ visual vectors. Although this improves compute efficiency (and allows us to train the model on just 2 A6000 GPUs), it results in the loss of some finegrained visual information. Scaling the model up to adopt pretraining objectives that encode more explicit information (e.g., longer sequences of ViT patches [4]) would be promising directions for future work.
>
>
> ### References
>
> [1] Koh, J.Y., Salakhutdinov, R. and Fried, D. “Grounding Language Models to Images for Multimodal Inputs and Outputs”. ICML, 2023.
>
> [2] Alayrac, Jean-Baptiste, et al. "Flamingo: a visual language model for few-shot learning." NeurIPS, 2022.
>
> [3] Aghajanyan, Armen, et al. "CM3: A causal masked multimodal model of the internet." arXiv preprint arXiv:2201.07520, 2022.
>
> [4] Liu, Haotian, et al. "Visual instruction tuning." arXiv preprint arXiv:2304.08485 (2023).

---

> > ### Comment · Reviewer_5sAf · 2023-08-18
> >
> > Thank you for the detailed rebuttal. Thanks for running the experiment without the retrieval loss to show that it is not bottlenecked by it! I believe the paper's results are very promising and adapting multimodal LLMs without too many resources is an exciting area. Hence, I maintain my score as a strong accept.

---

> > > ### Author Response · Authors · 2023-08-19
> > >
> > > Thank you for your helpful comments, and we are glad you liked the paper! We will also include the retrieval loss ablation results in the appendix of the next version of the paper. Please let us know if there are any further clarifications we can provide.

---

### Official Review · Reviewer_bnn7 · 2023-07-08

**Soundness:** 3 good
**Presentation:** 3 good
**Contribution:** 3 good
**Rating:** 5
**Confidence:** 4

**Summary:**

This paper proposes to use LLM to do image generation. Their approach consists of two stages of training.

In the first stage, they try to learn a linear layer to make VIT visual feature space is compatible with LLM space.

In the second stage, they learn r new tokens representing image. They hided states of these 'image token' are used to do retrieval or image generation. For the image generation, they train a GILLMapper to transfer these hidden states into CLIP text space (which is the input to the unet of SD) so that the transferred feature can be directly used as input the SD. They also found that simple design of the GILLMapper such as linear layer does not work well, thus they train a encoder -decoder like model (like a DETR)

**Strengths:**

The writing is clear and I like their idea and motivation in general.

**Weaknesses:**

1, I feel maybe it is not appropriate to claim their method can 'generate image'. The generation part is still from SD; what they are doing is to use a multi modal language model to combine both image and text information into CLIP text space. This can be reflected from their training: Neither stage1 nor stage2 actually involve image generation.

2, for table 1, I think current 1 caption case does not make scene, and maybe they can just skip this, as it is not using any previous information at all. For 5 caption case, I don't know how exactly they feed them (maybe just stack them?). I think a better way is to ask a LLM such as GPT4 to combine all captions and generate a better prompt.

3, for 5 captions and 4 images case in the table 1, the SD can also do this. A simple approach is to use image captioning model such as BLIP2, LLAVA etc to convert images as caption, and prompt GPT4 to generate the final caption by giving 5 captions and 4 images (in the captioning format)


**Questions:**

na

**Limitations:**

In my opinion, their comparison is not complete as they can have a simple extension (see weakness) so that it may potentially improve SD performance.

I will raise my score if they could conduct this experiment. If they are going to do the experiment, I hope they carefully prompt a LLM and choose different captioning models to evaluate.



------------------------------------------------------------------------------------------------------------
The authors address my major concerns in their rebuttal and I recommend them to add these stronger baselines in their final version.

W8 from reviewer QYwk seems a valid concern which I did not notice before. However, I am not very familiar with evaluation on text, thus I can not confidently evaluate their rebuttal in that question.

Overall, I am lean towards the accept, but will not be supervised if got rejected due to other weakness raised by other reviewers

---

> ### Author Rebuttal · Authors · 2023-08-07
>
> We thank the reviewer for their helpful comments. We are pleased that the reviewer found the writing clear, and that they liked the idea and motivation of the paper. We address specific queries below, and will incorporate all feedback.
>
> ## 1. Whether GILL Can ‘Generate Images’
>
> We consider GILL to be the combined visual encoder, LLM, and visual decoder (i.e., the SD generator), not just the LLM. One of our contributions is that this combined model can process interleaved image + text inputs, and generate text, generate images, and retrieve images, and interleave image and text in its outputs. Thus, we think that it is reasonable to claim that GILL can generate images, as this is one of its capabilities. If the reviewer has suggestions for specific lines in the paper that they believe could be re-written to be more accurate, we are also happy to consider rephrasing them.
>
>
>
> ## 2. 1 Caption Setting
>
> For the VIST evaluation, the single caption corresponds to the last image. Since the goal is to generate the last image, it is possible to produce a reasonable image given just this one caption. This allows us to compare what happens when the model has access to a single caption, for the last image, compared to the full sequence of captions, for all images, and it verifies that having access to the full context helps. For these reasons, we think that this is still a useful baseline to include.
>
> ## 3. 5 Captions + GPT Baseline
>
> The results in Tables 1 and 2 concatenate the 5 captions for both SD and our model. We also ran the experiment suggested by the reviewer, where we provided GPT-3.5 (turbo) with the 5 story captions, and asked it to generate a prompt.. We find that our GILL approach outperforms this baseline, see results below. (Below, in 4., we also find that our approach outperforms using GPT-4 as the LLM.) We prompt the LLM as follows:
>
> ```
> There are five images and five story parts that make up a full visual story. The five story parts are provided. Generate a caption that will be used as input to a text-to-image generation model to generate the last image in the sequence. The caption should relate to the full story sequence, but also describe the content that should be in the last image in the sequence.
>
> Story Part for Image 1: "my sister arrived early to help me with the family bar bq ."
> **<parts 2 to 4 omitted for brevity>**
> Story Part for Image 5: "we ended the day shooting off some fireworks ."
>
> Final image description for text-to-image generation:
> ```
>
> We find that this generally produces reasonable captions. For the above example, the completion by GPT-3.5 (turbo) is “a group of people gathered around a bonfire, watching fireworks light up the night sky”. We use this completion to generate an image with SD, and run the same VIST evaluation on it. The results are as follows:
>
> | Model      | CLIP Similarity ($\uparrow$) | LPIPS ($\downarrow$) |
> | ----------- | ----------- | ----------- |
> | Ours (5 captions + 4 images)                                 | **0.641**       | **0.693**       |
> | Ours (5 captions)                                 |  0.612      |  0.696       |
> | SD (5 captions, concatenated)         |  0.598        | 0.704      |
> | GPT-3.5 + SD  (5 captions)        | 0.605        | 0.707      |
>
>
> We see that while this improves over simply concatenating the captions, it still underperforms our method even when image information is not provided (despite GPT-3.5 using a much larger LLM). For reasons of cost, we do not evaluate GPT-4 in this setting, but do in 4. below.
>
>
> ## 4. Captioning Model + GPT + SD extension
>
> As suggested, we ran this baseline. We used various captioning model to generate captions for the 4 images, and similar to the above, prompted GPT-4 / GPT-3.5 as:
>
> ```
> There are five images and five story parts that make up a full visual story. The descriptions of the first four images and the five story parts are provided. Generate a caption that will be used as input to a text-to-image generation model to generate the last image in the sequence. The caption should relate to the full story sequence, but also describe the content that should be in the last image in the sequence.
>
> Story Part for Image 1: "my sister arrived early to help me with the family bar bq ."
> Image Description 1: "a woman sitting in the driver's seat of a car"
> **<parts 2 to 3 omitted for brevity>**
> Story Part for Image 4: "there was so much food and it was all delicious ."
> Image Description 4: "a person is holding a knife and a burger"
> Story Part for Image 5: "we ended the day shooting off some fireworks ."
>
> Final image description for text-to-image generation:
> ```
>
> Where the “Image Descriptions” are the outputs of an image captioning model. The results of this experiment on VIST are as follows (all models use 5 captions + 4 images):
>
> | Model      | CLIP Similarity ($\uparrow$) | LPIPS ($\downarrow$) |
> | ----------- | ----------- | ----------- |
> | Ours                                 | **0.641**       | **0.693**       |
> | BLIP-2 (OPT-6.7B) + GPT-3.5 + SD   | 0.620        | 0.705      |
> | BLIP-2 (T5-XL) + GPT-3.5 + SD        |  0.622        | 0.705      |
> | LLaVA (LLaMA-2-13B) + GPT-3.5 + SD        |  0.620        |  0.704      |
> | BLIP-2 (T5-XL) + GPT-4 + SD        |  0.630        |  0.700      |
>
>
> We find that this baseline improves results over the SD baseline (which does not use image captioning models), but the results are still worse compared to our model. This is despite our model being significantly smaller and finetuned with less data.
>
> Aside from these quantitative results, there are other benefits in using an approach such as ours. GILL generates latent features that are provided to the SD image generator, which bypasses the text bottleneck that models such as the GPT baseline above introduce. With our approach, there is no requirement to autoregressively generate text for the generator.
>
> We hope that this clarifies the reviewer’s queries on the suggested SD + GPT baselines.

---

### Author Rebuttal · Authors · 2023-08-07

We thank all reviewers for their valuable comments. We are glad that all reviewers appreciated the creativity and novelty of our method, and found it innovative and inspiring (reviewer Gk8n), appreciated its creative exploration (reviewer QYwk), ideas and motivation (reviewer bnn7), and recognized that it introduces a fundamental novelty in the capabilities of multimodal language models that other contemporary models lack (reviewer 5sAf).

We are pleased that the reviewers found the paper well-written (reviewers bnn7, Gk8n), and the results and applications impressive and very promising (reviewers Gk8n 5sAf). We are glad that the reviewers found our proposed modeling approaches insightful (reviewer Gk8n), and recognized its potential for multiple interesting follow-up works (reviewer 5sAf).

We address specific queries and run the suggested baselines in responses to individual reviewers below.

---

### Decision · Program_Chairs · 2023-09-21

**Decision:**

Accept (poster)

**Comment:**

This paper proposes to use LLM to do image generation. Their approach consists of two stages of training. (1) Stage-1: learning a linear layer to align VIT visual feature space with LLM word embedding space. (2) Stage-2: learning new tokens to represent the image. A GILLMapper is trained to transfer these hidden states into CLIP text space as the input to SD. Technique wise, a simple linear projection design of the GILLMapper does not work well, thus a encoder -decoder like model is considered.

All reviewers agree to accept the paper after the rebuttal. This is the first attempt to connect pre-trained image generation with open-source LLM, various technique details are well explored. This can be a valuable baseline for future exploration.